# *Doenjang* Ameliorates Diet-Induced Hyperlipidemia and Hepatic Oxidative Damage by Improving Lipid Metabolism, Oxidative Stress, and Inflammation in ICR Mice

**DOI:** 10.3390/foods13101471

**Published:** 2024-05-10

**Authors:** Olivet Chiamaka Edward, Do-Youn Jeong, Hee-Jong Yang, Anna Han, Youn-Soo Cha

**Affiliations:** 1Department of Food Science and Human Nutrition, Jeonbuk National University, Jeonju 54896, Republic of Korea; olivetedward@gmail.com (O.C.E.);; 2Microbial Institute for Fermentation Industry (MIFI), Sunchang 56048, Republic of Korea; 3K-Food Research Center, Jeonbuk National University, Jeonju 54896, Republic of Korea

**Keywords:** hyperlipidemia, hepatic oxidative damage, *Doenjang*, lipid metabolism, diet-induced hyperlipidemia

## Abstract

Hyperlipidemia, characterized by elevated cholesterol, lipids, and triglycerides in the bloodstream, is linked to hepatic oxidative damage. *Doenjang*, a traditional Korean condiment made from fermented soybeans, is known for its health benefits, yet its anti-hyperlipidemic effects remain understudied. Our study aimed to assess the hypolipidemic and hepatic protective effects of *Doenjang* on male ICR mice fed a high-fat cholesterol diet for 8 weeks. Mice were divided into three groups: the normal diet (ND), the high-fat cholesterol diet (HD), and the *Doenjang*-supplemented HD diet (DS) group. *Doenjang* supplementation significantly regulated total cholesterol, triglycerides, LDL cholesterol, and HDL cholesterol levels compared to the HD group. It also downregulated lipogenic genes, including PPARγ, FAS, and ACC, and positively influenced the cholesterol metabolism-related genes HMGCR and LXR. Moreover, *Doenjang* intake increased serum glutathione levels, activated oxidative stress defense genes (NRF2, SOD, GPx1, and CAT), positively modulated inflammation genes (NF-kB and IL6) in hepatic tissue, and reduced malondialdehyde levels. Our findings highlight the effectiveness of traditional *Doenjang* in preventing diet-induced hyperlipidemia and protecting against hepatic oxidative damage.

## 1. Introduction

Hyperlipidemia is a common medical condition characterized by elevated levels of lipids, notably cholesterol, such as low-density lipoprotein cholesterol (LDL-C) and triglycerides, in the bloodstream and is a significant risk factor for cardiovascular diseases such as atherosclerosis, coronary artery disease, stroke, and oxidative damage [1]. Effective management of hyperlipidemia is crucial for mitigating these associated risks. While conventional medical treatments like statins [2,3], a class of medications that inhibit cholesterol production in the liver, have demonstrated efficacy, concerns regarding adverse side effects persist [4], prompting interest in exploring alternative, natural, and dietary interventions. One such possible intervention is the potential use of *Doenjang*, a traditional Korean fermented soybean paste known for its various health benefits and potential for managing hyperlipidemia [5,6].

*Doenjang*, a traditional Korean condiment, is made from fermented soybeans and salt. It is a staple in Korean cuisine and is renowned for its rich umami flavor. Previous studies have highlighted its health benefits, such as antioxidant, anti-obesity, antimutagenic, and anti-cancer activities [6,7,8]. The fermentation process used in making *Doenjang* involves beneficial microorganisms such as *Aspergillus species* and *Bacillus subtilis*, which break down components of soybeans, potentially enhancing the bioavailability of essential amino acids, minerals, and vitamins [9,10]. This fermentation process may contribute to the observed health benefits of soy-based fermented foods, as fermented soybean products have demonstrated higher phenolic and protein contents along with greater antioxidant activity compared to their non-fermented counterparts [11]. Some studies suggest that consuming soy-based foods like *Doenjang* may positively influence lipid profiles by reducing low-density lipoprotein cholesterol (LDL-C) levels, often referred to as “bad” cholesterol [12,13]. In a hyperlipidemic state, lipogenic genes such as Sterol Regulatory Element-Binding Proteins (SREBPs), Fatty Acid Synthase (FAS), and Acetyl-CoA Carboxylase (ACC) are often overactivated, leading to increased synthesis of fatty acids and cholesterol [14]. Concurrently, oxidative stress genes such as Nuclear Factor Erythroid 2-Related Factor 2 (Nrf2), Superoxide Dismutase (SOD), Glutathione Peroxidase (GPx), and Catalase (CAT), when impaired, lead to decreased antioxidant defense and heightened oxidative damage [15,16]. These combined mechanisms promote lipid accumulation and oxidative stress, contributing to the pathogenesis of hyperlipidemia and its associated complications.

The relationship between *Doenjang* and hyperlipidemia is somewhat complex. The presence of bioactive compounds, including isoflavones and peptides, in *Doenjang* may help regulate cholesterol levels, as isoflavones have been known to reduce LDL-C levels and increase high-density lipoprotein cholesterol (HDL-C) [17], the “good” cholesterol. Furthermore, they enhance antioxidant properties, which help reduce oxidative stress in the body, which can contribute to the development of atherosclerosis [18]. However, while there have been studies on *Doenjang* with regards to obesity and oxidative stress [6,19,20], these studies focused on adipose tissues, were clinical trials, or possibly both. However, there is little to no research on the anti-hyperlipidemic properties of *Doenjang* or the effects on oxidative stress parameters in the liver in ICR mice. Therefore, our research aims to fully understand the mechanisms and efficacy of *Doenjang* in managing hyperlipidemia induced by a high-fat cholesterol diet, as well as potential health benefits regarding glucose metabolism and hepatic oxidative stress damage, which are accompanying risk factors associated with hyperlipidemia.

## 2. Materials and Methods

### 2.1. Animals and Diets

A *Doenjang* sample from the Sunchang region of the Republic of Korea was acquired from the Sunchang Microbial Institute for Fermented Foods, Republic of Korea. *Doenjang* is prepared using soybeans matured for 6 months by the traditional Korean fermentation process. The process involved steaming raw soybeans, which were then made into blocks called *meju*. Then *meju* is fermented with *Bacillus subtilis* and *Aspergillus oryzae* for a period of one month. After which, it was mixed with brine (saltwater) and further fermented for two additional months.

Male ICR mice (3-week-old) were purchased from Samtako Bio Korea (Gyeonggi-do, Republic of Korea). The use of ICR mice is an established model for the study of lipid metabolism. The animal diets were prepared following our laboratory protocol (Appendix A). After a two-week adaptation period, the mice were randomly assigned to three groups consisting of 10 mice per group: a normal diet (ND) group receiving a 10% kcal fat diet; a high-fat-cholesterol diet (HD) group receiving a 45% kcal fat diet; and a group fed an HD + 30% *Doenjang* (DS) for 8 weeks. The mice were kept in a controlled environment with a regulated temperature, following a 12-h light–dark cycle. Feed intake was recorded three times per week, and body weight and naso-anal length were recorded weekly. The energy intake, food efficiency ratio, and Lee index were calculated accordingly. Fasting glucose levels and systolic blood pressure values were measured monthly. The experimental protocol received approval from the Animal Care and Use Committee of the Jeonbuk National University (approval number: NON2023-087).

### 2.2. Serum and Tissue Sample Collection

At the end of the experimental period, mice underwent a 12-h fasting period and were sacrificed, and blood was collected from the retro-orbital vein using capillary tubes for serum extraction. After allowing the blood samples to stand for 30 min at room temperature, serum was obtained via centrifugation at 1500× *g* for 15 min at 4 °C and stored at −72 °C until analysis. Liver, pancreas, epididymal, and subcutaneous fat tissues were harvested, frozen in liquid nitrogen, and stored at −72 °C for subsequent analysis.

### 2.3. Lipid Profiles in the Liver and Serum

The levels of total cholesterol (TC) (AM202-K), triglyceride (TG) (AM157S-K), HDL-C (AM203-K), aspartate aminotransferase (AST; M101-k), and alanine aminotransferase (ALT; AM102-K) were measured enzymatically using commercially available kits (Asan Pharmaceutical Co., Seoul, Republic of Korea). LDL-C values were calculated using Friedewald’s equation [21], while very-low-density lipoprotein cholesterol (VLDL-C) values were calculated using the following equation: TG/5. The atherosclerosis index (AI) was determined using the following equation: TC/HDL-C. Hepatic TG and TC levels were analyzed following a previously established protocol [22]. 

### 2.4. Biochemical Analyses of Oxidative Damage and Insulin Levels

Commercial ELISA kits purchased from Elabscience Biotechnology Inc. (Houston, TX, USA) and Alpco (Salem, NH, USA) were used to measure serum levels of insulin (80-INSMSU-E01), malondialdehyde (MDA) (E-EL-0060), and glutathione (GSH) (E-EL-0026). The experiments were conducted in accordance with the instructions provided in their respective manuals. The Agribio online ELISA calculator tool was used for graph plotting and value calculation. HOMA-IR values were calculated using the following formula:

HOMA-IR = fasting insulin (microU/L) × fasting glucose (nmol/L)/22.5.



### 2.5. Histology Analyses

The liver, pancreas, epididymal, and subcutaneous fat tissues were fixed in 10% formalin. Hematoxylin and eosin (H&E) staining was performed by KP&T (Cheongju-si, Republic of Korea). The stained areas were viewed using an Axiophot Zeiss Z1 microscope (Carl Zeiss, Gottingen, Germany) at scales of 20 nm and 40 nm at Jeonbuk National University’s Center for University-Wide Research Facilities.

### 2.6. Total mRNA Isolation and qRT-PCR

A quantitative real-time polymerase chain reaction (qRT-PCR) was conducted on hepatic and pancreatic tissues to determine the expression of genes associated with lipid metabolism, glucose metabolism, inflammation, and oxidative stress, which are well-known markers of hyperlipidemia development. Total RNA was isolated using Trizol reagent (TakaRaBio, Shiga, Japan), and the RNA concentration and purity were assessed using a BioDrop μLITE spectrophotometer (Biodrop, Cambridge, UK). Subsequently, one nanogram of the isolated RNA was reverse transcribed into cDNA using Takara PrimeScript RT Master Mix (Tokyo, Japan). The levels of RNA expression were quantified via qRT-PCR using the SYBR Green Realtime PCR Master Mix (Enzynomics, Daejeon, Republic of Korea) and a 7500 Real-Time PCR system (Applied Biosystems, Foster City, CA, USA). The list of primers and primer sequences used in this study is listed in Appendix A.

### 2.7. Western Blotting Analysis

Immunoblot analysis was undertaken to assess the protein expression of genes related to lipid metabolism, cholesterol metabolism, and oxidative damage defense in liver tissue. ACC, P-ACC, AMP-activated protein kinase (AMPK), P-AMPK, SREBP1, Peroxisome proliferator-activated receptor alpha (PPARα), Peroxisome proliferator-activated receptor gamma (PPARγ), FAS, Fatty Acid-Binding Protein 4 (FABP4), SOD1, SOD2, CAT, GPx1, GPx2, GPx1 homotetramer, and β-ACTIN were purchased from Cell Signaling Technology (Beverly, MA, USA). Hepatic tissue was homogenized in lysis buffer and centrifuged, and the resulting supernatant was used to measure the protein concentration. Subsequently, all samples were standardized to an equal protein concentration, combined with 5× protein buffer, heated to 95 °C, and then placed on ice. Each sample was then subjected to electrophoresis on SDS–polyacrylamide gels, transferred to PVDF membranes (Merck, Burlington, MA, USA), and subjected to blotting with respective antibodies. Image processing and analysis were carried out using the KwikQuant Image Analyzer ver. 5.9.

### 2.8. Statistical Analysis

The results are expressed as means ± standard error (SE). A statistical analysis was performed using one-way ANOVA in SPSS (version 17.0; SPSS Inc., Chicago, IL, USA). Differences between means were evaluated using Duncan’s multiple-range test and an independent *t*-test, with statistical significance set at *p* < 0.05. 

## 3. Results

### 3.1. Doenjang Improves Food Efficiency Ratio, Inhibits Body Weight Gains, and Tissue Weight

In order to establish an isocaloric intake equivalent among all groups, the food intake significantly differed between the ND and the HD and CS groups; however, energy or caloric intake did not significantly differ between all groups (Figure 1A,B). The food efficiency ratio was significantly reduced in DS mice compared to HD mice, while the ND mice had a significantly lower FER (Figure 1C). Over the experimental period, the body weight gain of the HD mice significantly increased compared to the DS and ND mice, respectively. The final body weight of the mice in the HD group was significantly higher than that of the mice in the ND and DS groups. However, the final body weight of the mice in the ND group was significantly lower than that of the mice in the DS and HD groups (Figure 1D,E). To evaluate the degree of obesity, Lee’s index, which correlates with body fat composition, was calculated. The Lee’s index of the HD group was significantly higher than that of the other groups, while that of the DS group was markedly higher than that of the ND group (Figure 1F). The intake of *Doenjang* significantly reduced the liver, epididymal, and subcutaneous fat tissue weights compared to those of the HD group. However, ND had significantly lower tissue weights (Figure 1G). The relative tissue weights of the subcutaneous and epididymal fat tissues were similarly lower in the ND and DS groups compared to the HD group; however, the relative liver weight of the liver tissue was not significantly different between the DS and HD mice (Figure 1H). Histology captures of epididymal and subcutaneous tissues show that the adipocytes of the epididymal and subcutaneous adipose tissues were significantly larger and irregular in shape in the HD group than in the ND and DS groups, suggesting *Doenjang* supplementation significantly decreased mean adipocyte size relative to that of HD mice (Figure 1I,J). These findings suggest that *Doenjang* supplementation inhibits weight gain by reducing visceral abdominal fat and body and tissue weight and decreasing fat accumulation.

### 3.2. Doenjang Improves Lipid Profiles and Systolic Blood Pressure

In serum, the ND and DS groups had significantly lower serum TC levels than the HD group (Figure 2A). In addition, serum TG levels in DS mice were significantly lower than in both ND and HD mice (Figure 2B). Furthermore, the HD group had significantly lower HDL-C levels than the DS group. Notably, the DS group had the highest levels of HDL-C, which were markedly higher than those of the other groups (Figure 2C). The LDL-C levels were calculated using Friedewal’s equation. The HD group had significantly higher LDL-C levels than the DS and ND groups, with ND having the lowest levels among all groups in terms of LDL-C levels (Figure 2D). Calculated VLDL-C levels showed that the *Doenjang*-treated group was also significantly lower than the HD group (Figure 2E). The atherosclerosis index (AI), which is used to determine blood lipid levels and is commonly used as an optimal indicator of dyslipidemia and cardiovascular diseases, was calculated as LDL/HDL. The results showed the HD group to have the highest levels, followed by DS and ND in that order (Figure 2F). This was reflected in the systolic levels throughout the experiment period, where the systolic levels of the HD mice were markedly higher than those of the other groups between the 4th and 8th weeks (Figure 2G). Furthermore, the hepatic TG and TC levels were significantly lower in the DS group than in the HD group. The ND group had the significantly lowest hepatic TG and TC levels (Figure 2H,I). In addition, a high-fat cholesterol diet significantly increased the levels of serum AST and ALT in the HD group compared to the ND and DS groups, indicating liver injury (Figure 2J,K).

### 3.3. Doenjang Improves Glucose Metabolism and Pancreatic Health

At the end of the experiment period, the HD group had significantly higher fasting glucose levels than the other groups, whereas the ND group had significantly lower glucose levels (Figure 3A,B). The ND and CS groups also had significantly lower insulin levels than the HD group (Figure 3C). Insulin resistance, a phenomenon linked to obesity, was evaluated using the homeostasis model assessment of insulin resistance (HOMA-IR) index. The HOMA-IR index of the HD group was significantly higher than those of the ND and CS groups (Figure 3D). Histology captures of the pancreas, a major tissue involved in glucose regulation, show interlobular fat deposition (circled) in the pancreas of HD mice as well as irregular and damaged islets of Langerhans (Figure 3E), indicative of aberrant glucose metabolism. Therefore, we carried out RT-PCR analysis to confirm these histology data. Glut 4 and PGC-1α were upregulated in the HD mice (Figure 3F,G). PGC-1α acts as a selective repressor of NF-κB toward IL-6 in the pancreas. PGC-1α deficiency markedly enhanced NF-κB-mediated upregulation of Il6 in the pancreas in pancreatitis, leading to a severe inflammatory response. In addition, upregulation of the G6PD gene in pancreatic β-cells can induce β-cell dysregulation through ROS accumulation. Our findings show that G6P expression levels in the HD group were markedly upregulated compared to the ND and DS groups (Figure 3H). These findings suggest that *Doenjang* intake improves insulin sensitivity and, to some extent, modulates glucose metabolism and has the potential to protect against inflammation (pancreatitis), oxidative damage, and ROS accumulation in the pancreas, corroborating previous findings on the anti-diabetic properties of *Doenjang*.

### 3.4. Doenjang Protects against Oxidative Stress Damage and Hepatic Inflammation

Serum GSH levels were significantly lower and, conversely, MDA (an end product of lipid peroxidation) levels were significantly higher in HD mice compared to ND and DS mice (Figure 4A,B). Using RT-PCR, we determined a marked increase in the hepatic mRNA expression levels of the antioxidant enzyme precursor NRF2 in DS mice (Figure 4C), which was in line with the upregulation of the expression levels of the antioxidant enzymes SOD, GPx1, and CAT in the liver of DS mice compared to HD mice (Figure 4D–F). In addition, the hepatic inflammation markers NF-kB and IL6 were checked, as inflammation is closely associated with oxidative damage. DS mice were significantly lower than both HD and ND mice (Figure 4G,H).

### 3.5. Doenjang Improves Lipid Profiles and Hepatic Oxidative Damage

Histology capture of the hepatic tissue of HD mice showed a significant increase in lipid accumulation and deposition in the hepatic tissue compared to the ND and DS mice (Figure 5A). Using RT-PCR, we checked the mRNA expression levels of lipid metabolism-associated genes involved in triglyceride synthesis (SREBP1c, ACC, FAS, and PPARγ) and found them significantly upregulated in the high-fat diet-fed mice compared to the ND and DS mice, with DS mice having significantly lower levels of ACC, FAS, and PPARγ (Figure 5B–E). Gene expression levels of PPARα, the gene involved in reducing triglyceride level by regulation of energy homeostasis, and CPT1, a rate-limiting gene for fatty acid oxidation, were significantly increased in the DS mice compared to the HD mice (Figure 5F,G). Furthermore, the cholesterol synthesis-promoting HMGCR gene was downregulated in the DS group, while LXR was significantly upregulated compared to the HD group (Figure 5H,I).

Results of the Western blot of genes related to triglyceride synthesis, cholesterol metabolism, and oxidative damage in hepatic tissue (Figure 6A) showed an increased P-AMPK/AMPK ratio and P-ACC/ACC ratio in the DS group compared to the HD group; however, there was no significant difference between both groups (Figure 6B,C). The Western blot of genes related to triglyceride synthesis and cholesterol metabolism in liver tissue showed that the protein expression of SREBP1 was significantly lower in DS compared to HD, while PPARα protein expression was significantly increased. In addition, FAS and FABP4 were significantly downregulated in the DS group compared to the HD group (Figure 6D). The Western blot of genes related to oxidative damage in hepatic tissue showed upregulated protein expressions of the antioxidant defense genes SOD1, SOD2, CAT, and GPx1 homotetramer; however, there were no significant differences between the HD and DS groups (Figure 6E).

## 4. Discussion and Conclusions

Hyperlipidemia is associated with multiple factors, including glucose metabolism dysregulation, oxidative stress damage, and inflammation. Recently, research on hyperlipidemia, particularly focusing on its association with eating behavior, has been conducted, especially on high-fat diet-induced hyperlipidemia [23,24,25]. In addition, high-fat diet (HFD) consumption can induce the production of free radicals and lower antioxidant levels, resulting in a redox imbalance and oxidative stress. It has been reported that oxidative stress is closely related to various diseases, and consumption of HFD could induce oxidative stress and increase reactive oxygen species generation, thereby decreasing the endogenous antioxidant level [26,27,28,29]. Therefore, the objective of this study was to evaluate whether *Doenjang* could serve as a dietary prevention of HFD-induced hyperlipidemia and hepatic oxidative stress. 

Hyperlipidemia and obesity share intricate metabolic processes in vivo. Lee’s index, a marker reflecting the degree of obesity, was markedly reduced by supplementation with *Doenjang* in the DS mice but was higher in the HD mice, confirming the induction of obesity in the experimental mouse model. Subjects with obesity have enlarged adipocytes that release increased amounts of fatty acids into the bloodstream. This dysregulated lipid metabolism in obesity can contribute to the development of hyperlipidemia, characterized by elevated levels of lipids such as cholesterol and triglycerides in the bloodstream. The excess fatty acids released from adipose tissue can lead to increased synthesis and secretion of triglyceride-rich very-low-density lipoproteins (VLDLs) by the liver. The findings of this current study indicated that *Doenjang* had a notable impact on reducing obesity caused by a high-fat diet, as evidenced by the significant decrease in body weight, epididymal and subcutaneous fat tissue weights, as well as significantly enlarged adipocytes in HD mice compared to DS mice in the H&E histology captures (Figure 1D,E,G–J). In addition, there were no significant differences in the daily caloric intake among the three groups (ND, HD, and DS), indicating that the weight gain of the mice was not caused by the differences in food intake. As aforementioned, high-fat diet consumption induces obesity and concurrently hyperlipidemia, leading to increased serum TC, TG, LDL-C, and VLDL-C levels, as well as hepatic TC and TG levels and the substantial accumulation of fat in the liver [30,31]. Our findings showed that the HD mice had significantly increased levels of TC, TG, LDL-C, and VLDL-C in the serum and/or liver, indicative of hyperlipidemia. These lipid levels were markedly reduced in the DS mice, which was consistent with a previous clinical study [31]. Furthermore, *Doenjang* intake increased HDL-C levels in the DS mice compared to the HD mice. These results suggest that *Doenjang* has a hypolipidemic effect and can regulate serum and liver lipid profiles.

The liver performs a vital role in the body, including the synthesis, metabolism, and storage of lipids [32]. Lipid synthesis is primarily controlled by two key enzymes that limit the rate of synthesis: FAS and ACC. FAS is governed by PPARγ’s positive feedback loop, while ACC is regulated by PPARγ’s negative feedback regulation. Activation of PPARγ triggers a heightened expression of proteins related to fatty acid transport and an increase in cellular triglyceride synthesis [33]. In addition, activation of AMPK increases PPARα expression, which suppresses the expression of ACC and concurrently boosts the expression of CPT1. PPARα, a nuclear transcription factor, plays an important role in the regulation of lipid metabolism, and it induces fatty acid transport and oxidation. Carnitine palmitoyltransferase 1a (CPT1a) is a downstream target gene of PPARα that mediates the entry of acyl into the mitochondrial matrix and is a key rate-limiting enzyme in fatty acid β-oxidation. As a result, there is a rise in the breakdown of lipids through fatty acid beta-oxidation in hepatic cells [33,34]. Therefore, to investigate the potential of the lipid-lowering effects of *Doenjang* in high-fat diet-induced hyperlipidemic mice, mRNA and protein transcription levels of PPARα and CPT1a are important markers [35]. Our study findings suggest that *Doenjang* supplementation can lower the expression of genes related to fat formation or lipogenesis in the liver of mice on a high-fat diet, including PPARγ, SREBP1c, FAS, and ACC, while increasing PPARα and CPT1 levels (Figure 5A–G and Figure 6A–D). 

The link between hyperlipidemia, lipogenesis, and hepatic oxidative stress involves a complex interplay of molecular pathways and metabolic processes. Elevated levels of lipids, such as cholesterol and triglycerides, in hyperlipidemia can lead to increased oxidative stress within cells and tissues. Excess lipids can promote the production of reactive oxygen species (ROS) through processes such as lipid peroxidation and mitochondrial dysfunction. ROS, in turn, can cause oxidative damage to cellular components such as lipids, proteins, and DNA. This oxidative stress not only exacerbates lipid metabolism dysregulation but also contributes to endothelial dysfunction, atherosclerosis, and other cardiovascular complications associated with hyperlipidemia. Oxidative stress due to an imbalance between the generation of ROS and endogenous antioxidant systems can promote hyperlipidemia, lipid peroxidation, and lipogenesis, while hyperlipidemia can further exacerbate hepatic lipid accumulation, dysregulate lipogenic pathways, and activate transcription factors such as sterol regulatory element-binding proteins (SREBPs), which are key regulators of lipogenesis. This, in turn, is involved in the etiology of several chronic diseases, such as cardiovascular diseases, diabetes, obesity, and cancer [36]. The levels of MDA are used as a biomarker of lipid peroxidation and oxidative stress [37]. As presented in Figure 4B, *Doenjang* consumption significantly decreased serum MDA levels by 52.5% in comparison with the HD group. Previous studies with Korean fermented foods such as kimchi [38] have reported that traditional Korean fermented foods could significantly reduce markers of oxidative stress. Antioxidants attenuate or inhibit the oxidation of lipids or other biomolecules and, thus, prevent or repair the damage to body cells due to oxidation by free radical species. Free radical species can be neutralized through dismutation or reduction by endogenous antioxidants like GSH, SOD, CAT, and GPx [39]. Our study showed that serum GSH levels, antioxidant enzymes (NRF2, SOD, CAT, and GPx), mRNA, and protein transcription/expression levels were significantly increased in the DS mice compared to the HD mice in the hepatic tissue. Therefore, our findings suggest that *Doenjang* possesses antioxidant properties and prevents hepatic oxidative damage, as evidenced by the lower levels of MDA (an oxidative stress end-product) and increased levels of antioxidant defense enzymes in the DS mice compared to the HD mice.

Oxidative stress is, in addition, a potent inducer of inflammation through the activation of various signaling pathways and transcription factors, including nuclear factor kappa B (NF-kB). ROS can directly activate NF-kB, a key regulator of inflammatory gene expression, leading to the production of pro-inflammatory cytokines such as interleukin-6 (IL-6) [40]. In the present study, we noticed that significantly higher levels of NF-kB and IL-6, the inflammation cytokines, were present in the hepatic tissue of the HD mice compared to the DS mice. The increase in these inflammatory cytokines might interfere with insulin action by suppressing insulin signal transduction [41]. This prompted our investigation of glucose metabolism in the serum and pancreatic tissue, and our results indicated that HD had significantly higher glucose and insulin levels as well as a higher HOMA IR index, indicating the onset of a hyperglycemic state (Figure 3A–D). A fatty pancreas is associated with abdominal obesity, insulin resistance, T2DM, dyslipidemia, arterial hypertension, and metabolic syndrome. This finding was supported by the improvement in histopathology of the pancreas in the DS mice, while the HD mice had fat deposits in their pancreatic tissue (Figure 3E). In addition, we recorded significantly elevated Glut4 mRNA expression in the pancreas, which may signify an adaptive response to increased glucose levels or insulin demand. This upregulation could facilitate enhanced glucose uptake into pancreatic β-cells, leading to improved insulin secretion in response to elevated blood glucose levels. Therefore, increased pancreatic Glut4 mRNA expression may indicate improved pancreatic function and glucose homeostasis, potentially serving as a marker for enhanced pancreatic β-cell function and insulin sensitivity. Furthermore, PGC-1α, a major regulator of mitochondrial function and energy metabolism, whose upregulation of PGC-1alpha expression promotes the transcription of genes involved in mitochondrial biogenesis, oxidative phosphorylation, and fatty acid oxidation, was significantly increased by *Doenjang* supplementation. Therefore, increased PGC-1alpha expression in the pancreas suggests an adaptive response to metabolic demands, potentially improving cellular energy production and oxidative stress defense mechanisms. Thus, *Doenjang* shows potential to correct oxidative damage and inflammation and may increase insulin signal transduction and improve pancreatic health, glucose metabolism, and insulin sensitivity in DS mice by inhibiting NF-kB activation and decreasing IL-6.

In conclusion, our study suggests that *Doenjang* possesses hypolipidemic properties and can improve hyperglycemia, alleviate oxidative stress and inflammation, and ameliorate dyslipidemia, thus providing preclinical evidence for *Doenjang* as a potential functional food for the management of hyperlipidemia. 

## Figures and Tables

**Figure 1 foods-13-01471-f001:**
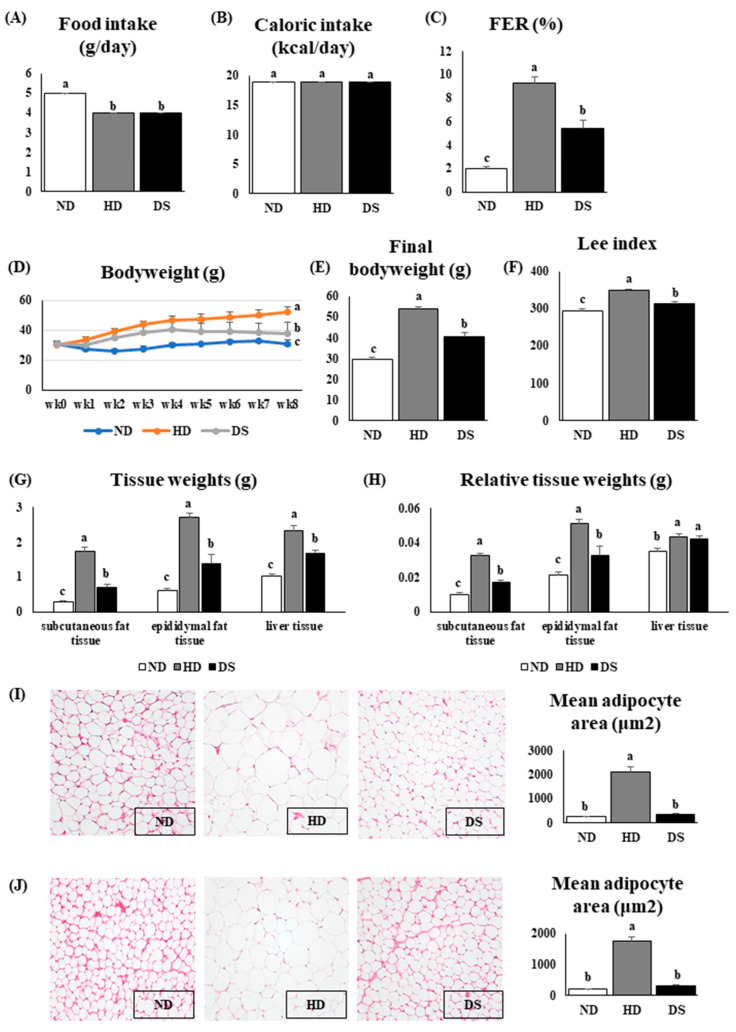
Effect of *Doenjang* intake on feed intake, body and tissue weights, and adipocyte morphology. (**A**) Daily food intake; (**B**) daily caloric intake; (**C**) food efficiency ratio; (**D**) body weight changes over the experiment period; (**E**) final body weight; (**F**) Lee index; (**G**) tissue weights (g) (subcutaneous fat tissue, epididymal fat tissue, and liver tissue weights); (**H**) relative tissue weight (g) (subcutaneous fat tissue, epididymal fat tissue, and liver tissue relative weights); (**I**) histological captures of epididymal fat tissue at a scale of 40 μm and the mean adipocyte area; (**J**) histological captures of subcutaneous fat tissue at a scale of 40 μm and the mean adipocyte area. Each bar represents the mean ± SE. A one-way ANOVA was used to find the statistical significance at *p* < 0.05. Bars with different upper scripts differ significantly from each other.

**Figure 2 foods-13-01471-f002:**
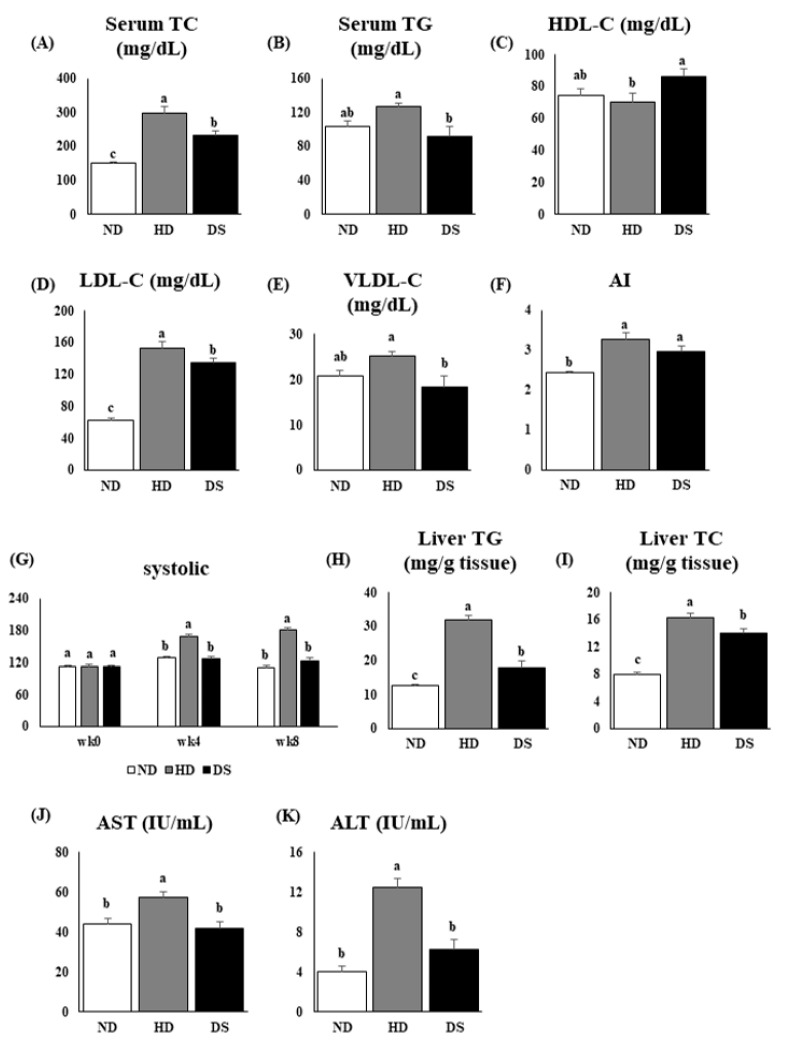
Effect of *Doenjang* on serum and hepatic lipid profiles and systolic blood pressure. (**A**) Serum total cholesterol levels; (**B**) serum total triglyceride levels; (**C**) high-density lipoprotein-cholesterol levels; (**D**) low-density lipoprotein-cholesterol levels; (**E**) very-low-density lipoprotein-cholesterol levels; (**F**) atherosclerosis index; (**G**) systolic blood pressure over the experiment period; (**H**) hepatic total triglyceride levels; (**I**) hepatic total cholesterol levels; (**J**) serum AST levels; (**K**) serum ALT levels. Each bar represents the mean ± SE. A one-way ANOVA was used to find statistical significance at *p* < 0.05. Bars with different upper scripts differ significantly from each other.

**Figure 3 foods-13-01471-f003:**
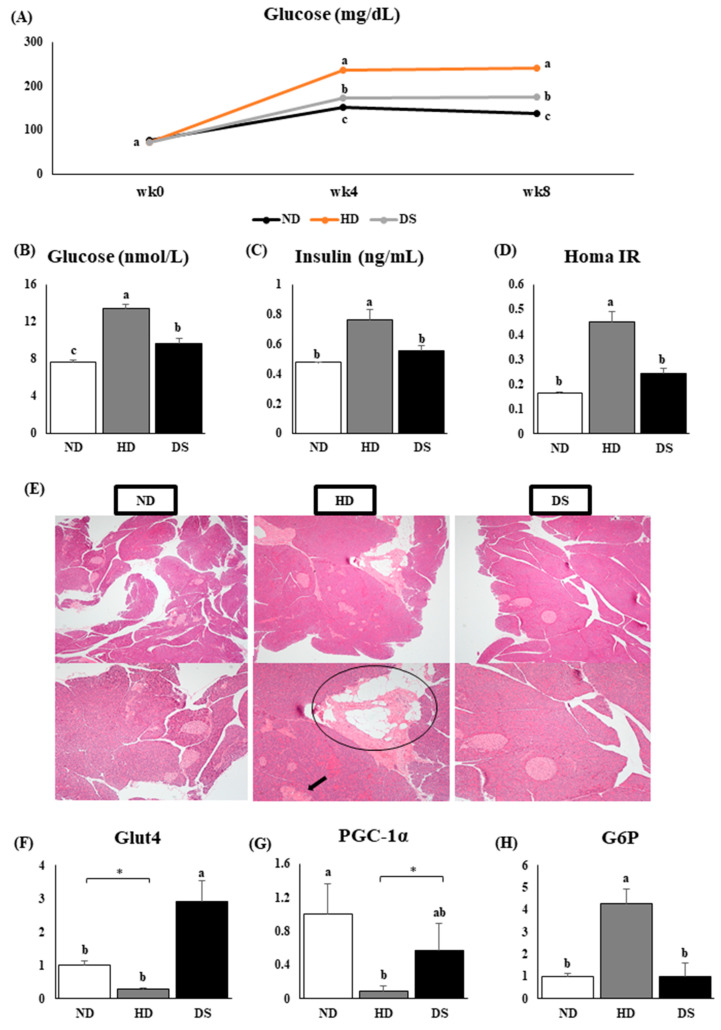
Effect of *Doenjang* on glucose metabolism, pancreatic histology, and mRNA expression of genes in pancreatic tissue. (**A**) Fasting glucose levels over the experiment period; (**B**) final fasting glucose levels in nmol/L; (**C**) serum insulin levels; (**D**) HOMA IR levels; (**E**) histological captures of pancreatic tissue at a scale of 20 μm and 40 μm (circled area shows fat deposits, arrow shows abnormal islets of Langerhans); (**F**) mRNA expression level of pancreatic Glut4; (**G**) mRNA expression level of pancreatic PGC-1α; (**H**) mRNA expression level of pancreatic G6P. Each bar represents the mean ± SE. A one-way ANOVA was used to find statistical significance at *p* < 0.05. * indicates a significant difference according to an independent *t*-test at *p* < 0.05. Bars with different upper scripts differ significantly from each other.

**Figure 4 foods-13-01471-f004:**
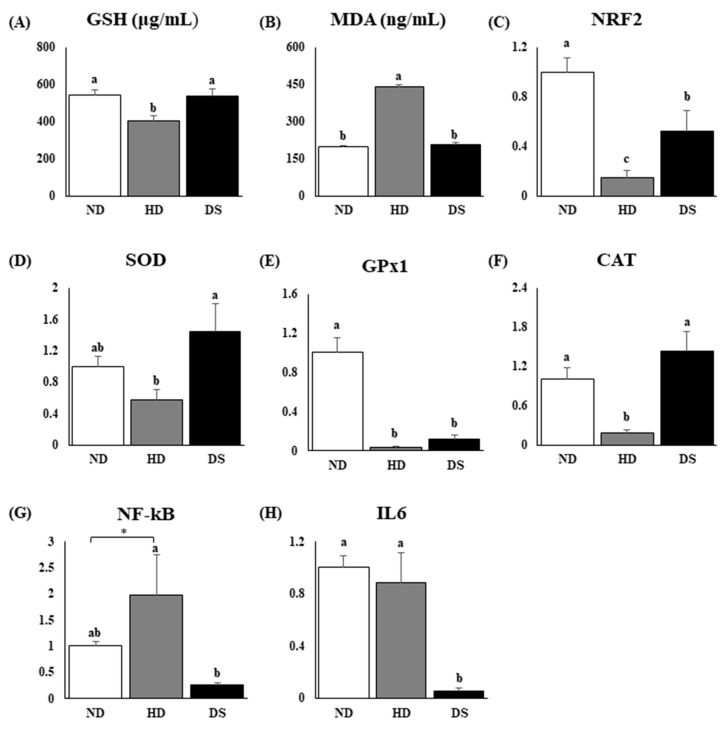
Effect of *Doenjang* on oxidative stress parameters and mRNA expression levels relative to beta-actin of oxidative stress and inflammation-related genes and enzymes in hepatic tissue. (**A**) Serum glutathione (GSH) levels; (**B**) serum malondialdehyde (MDA) levels; (**C**) mRNA expression level of NRF2; (**D**) mRNA expression level of SOD; (**E**) mRNA expression level of GPx1; (**F**) mRNA expression level of CAT; (**G**) mRNA expression level of NF-kB; (**H**) mRNA expression level of IL6. Each bar represents the mean ± SE. A one-way ANOVA was used to find statistical significance at *p* < 0.05. * indicates a significant difference according to an independent *t*-test at *p* < 0.05. Bars with different upper scripts differ significantly from each other.

**Figure 5 foods-13-01471-f005:**
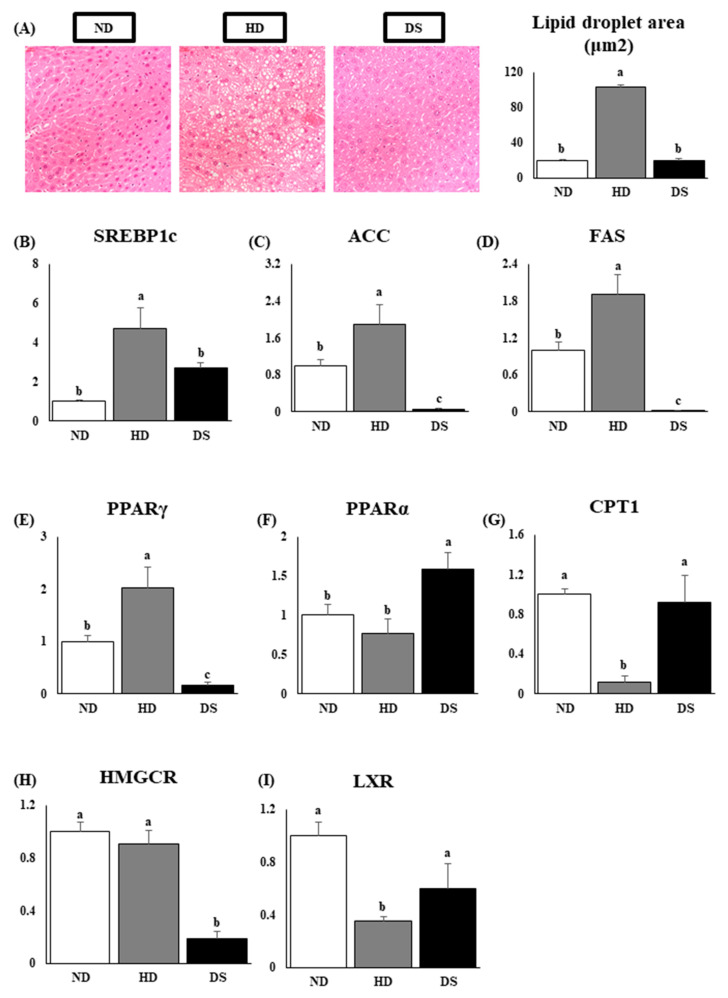
Effect of *Doenjang* on hepatic histology and mRNA expression levels relative to beta-actin of lipid, triglyceride synthesis, and cholesterol metabolism-related genes and enzymes in hepatic tissue. (**A**) Histological captures of liver tissue at a scale of 40 μm and hepatic lipid droplet area; (**B**) mRNA expression level of SREBP1c; (**C**) mRNA expression level of ACC; (**D**) mRNA expression level of FAS; (**E**) mRNA expression level of PPARγ; (**F**) mRNA expression level of PPARα; (**G**) mRNA expression level of CPT1; (**H**) mRNA expression level of HMGCR; (**I**) mRNA expression level of LXR. Each bar represents the mean ± SE. A one-way ANOVA was used to find statistical significance at *p* < 0.05. Bars with different upper scripts differ significantly from each other.

**Figure 6 foods-13-01471-f006:**
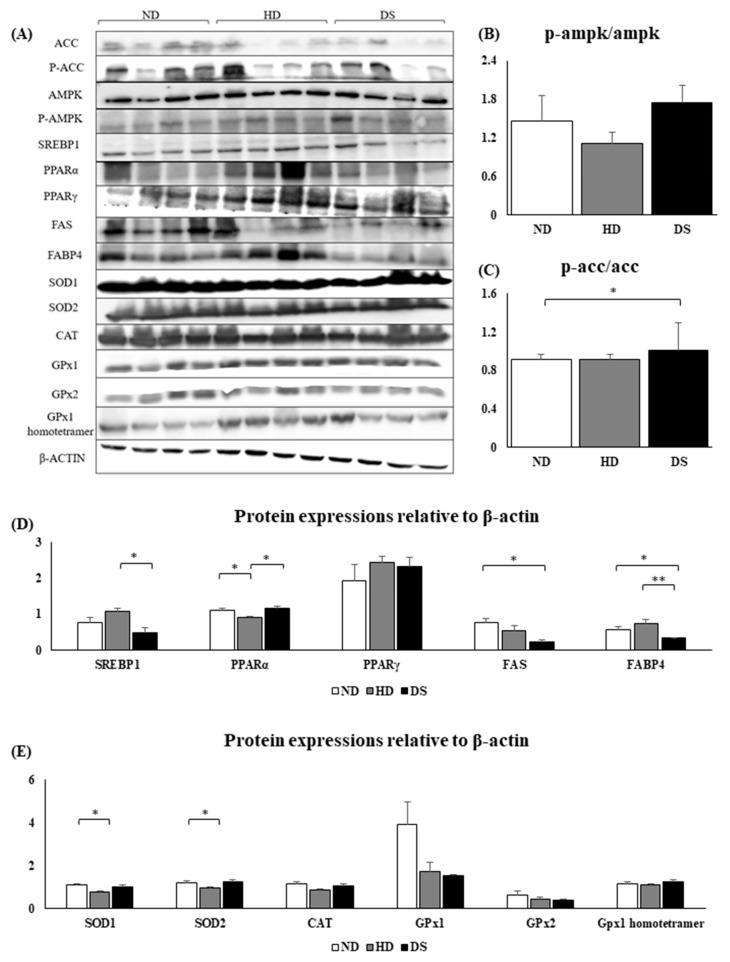
Effect of *Doenjang* on hepatic protein expressions of lipid metabolism, cholesterol metabolism, and oxidative damage defense-related genes. (**A**) Western blot images of investigated genes; (**B**) protein expression of p-AMPK/AMPK; (**C**) protein expression of p-acc/acc; (**D**) protein expressions of SREBP1, PPARα, PPARγ, FAS, and FABP4 relative to β-actin; (**E**) protein expressions of SOD1, SOD2, CAT, GPx1, GPX2, and GPx1 homotetramer form relative to β-actin. Each bar represents the mean ± SE. A one-way ANOVA was used to find statistical significance at *p* < 0.05. * indicates a significant difference according to an independent *t*-test at *p* < 0.05, ** indicates a significant difference according to an independent *t*-test at *p* < 0.01.

## Data Availability

The original contributions presented in the study are included in the article/Appendix A, further inquiries can be directed to the corresponding author.

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
