# Peer review of "Doenjang Ameliorates Diet-Induced Hyperlipidemia and Hepatic Oxidative Damage by Improving Lipid Metabolism, Oxidative Stress, and Inflammation in ICR Mice"

_foods, 2024, doi:10.3390/foods13101471_

Round 1

Reviewer 1 Report

Comments and Suggestions for Authors

The topic is interesting, the experiment was well-designed, and the manuscript is fairly written. However, some things have to be corrected to make the manuscript suitable for publishing. Here are my comments:

1.     The title needs to change to include the lipogenic and oxidative stress defense genes. This would be more precise and interesting and would also relate to the aim of this study, which is to fully understand the mechanisms and efficacy of doenjang in managing hyperlipidemia induced by a high-fat cholesterol diet.

2.     The abstract needs to add more detail on significant discoveries of lipogenic genes and the efficacy of doenjang on oxidative stress genes. For example, These results suggested that doenjang may modulate lipid metabolism via regulating PPARα and CPT1a mRNA expressions inhibiting lipogenesis in liver (Line 364-365)

3.     The introduction needs more mechanisms of lipogenic genes and oxidative stress genes related to this study.

4.     The methods need to be clear and follow-upable. Many results aren’t provided or described in the methods, and they need to cover all results, such as AST, ALT, Glucose, Insulin, HOMA-IR, GLUT4, G6P, NRF2, IL6, NF-kB, etc. The methods to support results in 3.4. Doenjang protects against oxidative stress damage and hepatic inflammation and needs to be described in methods.

5.     The results are well presented. However, the raw data of RT-PCR of all gene expression needs to be added along with Graphs in the main manuscript or supplementary file. Fig 6 (A) needs to improve the label of some protein names more clearly.

General comments;

-Since you abbreviated low-density lipoprotein cholesterol (LDL-C)” in line 29, and high-density lipoprotein cholesterol (HDL-C)” in line 55, there is no need to abbreviate the whole manuscript again (like lines 50/106 or 104).

-Must need to abbreviation VLDL-C, AST, ALT, SREBP1 (line 129), PPARα (not in line 352), PPARγ, CPT1 (line 266),

-Need ref citation for lines 70, 212, 355

-Please correct line 341,higher levels of nfkb,to higher levels of NF-kB.” Also, please correct line 264 (Fig 5B-D).

-Remove the numbering from References” (line 401). The reference style and symbol (μ, %) should follow the journal guidelines

Comments on the Quality of English Language

Moderate editing of English language required

Author Response

Dear Editor and Reviewers,

Thank you for giving us the opportunity to submit a revised draft of our manuscript with reference number foods-2960176 to your esteemed journal.

We appreciate the time and effort that you and the reviewers have dedicated to providing your valuable feedback on our manuscript. We are grateful to the reviewers for their insightful comments on our paper. We have been able to incorporate changes to reflect most of the suggestions provided by the reviewers. We have highlighted the changes in yellow within the manuscript.

(A) Respond to Reviewer’s Comments

  1. The title needs to change to include the lipogenic and oxidative stress defense genes. This would be more precise and interesting and would also relate to the aim of this study, which is to fully understand the mechanisms and efficacy of doenjang in managing hyperlipidemia induced by a high-fat cholesterol diet.

Response: We thank the reviewer for this insightful comment. We agree with this comment therefore we have followed the suggestion and changed the title to reflect the reviewer’s suggestion. The title is now ‘Doenjang Ameliorates Diet-Induced Hyperlipidemia and Hepatic Oxidative Damage by Improving Lipid Metabolism, Oxidative Stress and Inflammation in ICR Mice’.

  1. The abstract needs to add more detail on significant discoveries of lipogenic genes and the efficacy of doenjang on oxidative stress genes. For example, these results suggested that doenjang may modulate lipid metabolism via regulating PPARα and CPT1a mRNA expressions inhibiting lipogenesis in liver (Line 364-365)

Response: We appreciate the reviewer’s suggestions and have added additional information accordingly. “It also significantly downregulated lipogenic genes including PPARγ, FAS and ACC and positively influenced cholesterol metabolism-related genes HMGCR and LXR” and “Doenjang intake increased serum glutathione levels, activated oxidative stress defense genes; NRF2, SOD, GPx1 and CAT and positively modulated inflammation genes; NF-kB and IL6 in hepatic tissue and…”

  1. The introduction needs more mechanisms of lipogenic genes and oxidative stress genes related to this study.

Response: We have added some additional information regarding the mechanisms of lipogenic and oxidative stress genes as related to the current study in the introduction.

“In a hyperlipidemic state, lipogenic genes such as Sterol Regulatory Element-Binding Proteins (SREBPs), Fatty Acid Synthase (FAS), and Acetyl-CoA Carboxylase (ACC) are often overactivated, leading to increased synthesis of fatty acids and cholesterol [14]. Concurrently, oxidative stress genes such as Nuclear Factor Erythroid 2-Related Factor 2 (Nrf2), Superoxide Dismutase (SOD), Glutathione Peroxidase (GPx), and Catalase, when impaired, lead to decreased antioxidant defense and heightened oxidative damage [15,16]. These combined mechanisms promote lipid accumulation and oxidative stress, contributing to the pathogenesis of hyperlipidemia and its associated complications.”  Line 54-62

  1. The methods need to be clear and follow-upable. Many results aren’t provided or described in the methods, and they need to cover all results, such as AST, ALT, Glucose, Insulin, HOMA-IR, GLUT4, G6P, NRF2, IL6, NF-kB, etc. The methods to support results in 3.4. Doenjang protects against oxidative stress damage and hepatic inflammation and needs to be described in methods.

Response: We appreciate the reviewer’s keen insight and have corrected this oversight in the methods section accordingly to reflect all results.

  1. The results are well presented. However, the raw data of RT-PCR of all gene expression needs to be added along with Graphs in the main manuscript or supplementary file. Fig 6 (A) needs to improve the label of some protein names more clearly.

Response: We appreciate the comment. However, as regard to the raw data of the RT-PCR experiments, we are unsure of what exactly is being requested so we will submit an excel file of the raw values gotten during the experiment depicting the calculations of how we got the fold change values relative to beta-actin.

General comments;

-Since you abbreviated “low-density lipoprotein cholesterol (LDL-C)” in line 29, and “high-density lipoprotein cholesterol (HDL-C)” in line 55, there is no need to abbreviate the whole manuscript again (like lines 50/106 or 104).

-Must need to abbreviation VLDL-C, AST, ALT, SREBP1 (line 129), PPARα (not in line 352), PPARγ, CPT1 (line 266),

-Need ref citation for lines 70, 212, 355

-Please correct line 341, “higher levels of nfkb,” to “higher levels of NF-kB.” Also, please correct line 264 (Fig 5B-D).

-Remove the numbering from “References” (line 401). The reference style and symbol (μ, %) should follow the journal guidelines.

Response: We have amended these listed corrections and these changes are indicated in yellow throughout the manuscript. Thank you.

Additional clarifications

We have revised our manuscript to address the various problems brought up by the esteemed reviewers and hope we have been able to resolve the reviewers’ concerns and that our manuscript is now acceptable to be published in your esteemed journal.

In addition to the above comments, all spelling and grammatical errors pointed out by the reviewers have been corrected.

Sincerely,

Edward Olivet Chiamaka

First author

2024-04-11

Reviewer 2 Report

Comments and Suggestions for Authors

In this study, the authors examined the effects of doenjang, made from fermented soybean and salt, in a mouse obesity model fed a high-fat diet. Although doenjang (DS) has been shown to have strong effects, there are several major problems.

[Major points]

1.  Introduction: In addition to the effects of DS on obesity and blood lipids, antioxidant and anti-inflammatory effects have also been reported (For example: Nam et al. Nutr Res Pract. 2015). It is necessary to cite these documents in the text and clearly state how the present study differs from these studies.

2. Percentage of DS added to the diet:  According to the description of the method (lines 79-80), is the percentage of DS added to a high fat diet 30%? If true, this addition is too much. It is usually no more than a few percent. The caloric intake is unchanged in the DS group compared to the high-fat group, but the intake of essential nutrients such as vitamins seems to be deficient in the DS group. Therefore, the results of this study may not be an accurate assessment of DS

3.   Discussion: In the discussion section, results are only reiterated and not discussed or explained in depth. A more in-depth description is needed. For example, in this study, many indices were measured, including parameters of lipid synthesis and metabolism, oxidative stress, and inflammation, and DS affects most of these. What exactly is the mechanism by which DS suppressed obesity, diabetes, hyperlipidemia, fatty liver, etc. is largely undiscussed. These should be explained in detail in Discussion.

4.    Title: Parameters of lipid synthesis and metabolism, oxidative stress, and inflammation have been significantly altered, and in addition to lipid-lowering effects, suppression of obesity and diabetes has been observed. Therefore, the title of this paper, Inhibition of Hyperlipidemia and Oxidative Stress, needs to be revised.

5.   Experimental Data: DS altered several indices very strongly. For example, in Figure 4, DS intensely inhibited NF-kB and IL-6 mRNA expression to below normal levels. In Figure 5, it also suppressed ACC and FAS expression, which are involved in lipid synthesis and metabolism, to almost zero. Given these data, such changes observed with high doses of DS should be considered as side effects rather than an effect. Alternatively, it may be related to the lack of essential nutrients mentioned earlier. In Figure 3, DS markedly increased pancreatic Glut4 mRNA expression. The authors need to explain the reason for this. In this study, DS was evaluated in high-fat-fed mice, but what would be expected if DS were given to normal mice? If necessary, the study should be conducted in normal mice.

6.   Ingredient composition of DSAs noted above, there is no clear explanation of which parts of the body DS acts on and how it exhibits obesity and lipid-lowering effects, but another problem is that there is no description of what ingredients are in DS. It appears to be a fermented soybean product with salt added as an ingredient, but it needs to be clarified what ingredients are responsible for these effects. What is the substance that is acting? Is it acting in the intestinal tract or is it absorbed and acting? How much salt does DS contain, and there is a concern that salt may have a significant effect when DS is given in large doses?

[Minor points]

line 79: HFC may be a mistake for HD

line 79-80: What is unchang region? Is DS a mistake?

Overall, this paper does not reach the level of acceptance, including the content of the discussion. The most essential problem is the content of the experiment (ratio of DS added to feed). For the paper to be accepted, these issues need to be logically explained and resolved. 

Comments on the Quality of English Language

As a non-native speaker, the quality of the English in this manuscript is at an avarage level.

Author Response

Dear Editor and Reviewers,

Thank you for giving us the opportunity to submit a revised draft of our manuscript with reference number foods-2960176 to your esteemed journal.

(B) Respond to Reviewer's Comments

In this study, the authors examined the effects of doenjang, made from fermented soybean and salt, in a mouse obesity model fed a high-fat diet. Although doenjang (DS) has been shown to have strong effects, there are several major problems.

[Major points]

  1. Introduction: In addition to the effects of DS on obesity and blood lipids, antioxidant and anti-inflammatory effects have also been reported (For example: Nam et al. Nutr Res Pract. 2015). It is necessary to cite these documents in the text and clearly state how the present study differs from these studies.

Response: We thank you for your insightful comment, as mentioned in the introduction section, we acknowledge there have been studies on doenjang focusing on various health conditions. The reviewer aforementioned study for example is focused on obesity and oxidative stress in adipose tissue. Our study is focused on lipid metabolism focusing on hepatic tissue. This is because the liver is the major tissue involved in lipid metabolism. In addition, since we know oxidative stress is an accompanying factor of dyslipidemia, we also focused on assessing the effect doenjang had on oxidative stress parameters in the liver. We have now added this information to the introduction part of the manuscript to state how the present study differs from other studies. Line 68-72

  1. Percentage of DS added to the diet: According to the description of the method (lines 79-80), is the percentage of DS added to a high fat diet 30%? If true, this addition is too much. It is usually no more than a few percent. The caloric intake is unchanged in the DS group compared to the high-fat group, but the intake of essential nutrients such as vitamins seems to be deficient in the DS group. Therefore, the results of this study may not be an accurate assessment of DS

Response: Thank you for the question. We arrived at the decision to add 30% doenjang to HD diet, based on previous research we have undertaken as well as some other published research regarding supplementation of traditional fermented foods (Edward OC, Lee EJ, Han A, et al. Gochujang Consumption Prevents Metabolic Syndrome in a High-Fat Diet Induced Obese Mouse Model. J Med Food. 2023 Apr;26(4):244-254.) Kim J, Choi JN, Choi JH, Cha YS, Muthaiya MJ, Lee CH. Effect of fermented soybean product (Cheonggukjang) intake on metabolic parameters in mice fed a high-fat diet. Mol Nutr Food Res. 2013 Oct;57(10):1886-91. doi: 10.1002/mnfr.201200700. Epub 2013 Apr 23. PMID: 23609950 and other Korean based articles. As regards to the caloric intake and the concern of essential nutrients and vitamins we have attached a supplementary file of the diet compositions used in the present study, which shows in detail that both HD and DS diet composition have similar amounts of all essential macro and micro nutrients including minerals and vitamins. We have used this diet compositions in previous published research involving traditional Korean fermented foods. Please refer to supplementary table 1: diet composition.

Table S1: Diet composition

ND

HD

DS

casein

117.5

117.5

105.7163

L-Cystine

2.75

2.75

2.75

corn starch

315

106

91.87596

maltodextrin 10

50

50

50

Sucrose

50

50

50

lard

16.5

110

107.6085

soybean oil

10

10

10

Cholesterol

0

1

1

Cellulose

25

25

25

mineral mix

17.5

17.5

17.5

Calcium Carbonate

15

15

15

Vitamin Mix

5

5

5

Choline bitartrate

1.25

1.25

1.25

된장

0

0

102.2

total(g)

625.5

511

584.9008

kcal

4759

4770

4770

kcal/g

3.804157

4.667319

4.667319

Note: The carbohydrate, protein and fat content of the Doenjang sample was taken into consideration during formulation and were adjusted accordingly to result in similar ratio of all macronutrients.

  1. Discussion: In the discussion section, results are only reiterated and not discussed or explained in depth. A more in-depth description is needed. For example, in this study, many indices were measured, including parameters of lipid synthesis and metabolism, oxidative stress, and inflammation, and DS affects most of these. What exactly is the mechanism by which DS suppressed obesity, diabetes, hyperlipidemia, fatty liver, etc. is largely undiscussed. These should be explained in detail in Discussion.

Response: We appreciate your comment, and have revised the entire discussion section to reflect these points for easier readability, and more in-depth discussions about the main results and their implications, creating a well-rounded discussion of the current study.

  1. Title: Parameters of lipid synthesis and metabolism, oxidative stress, and inflammation have been significantly altered, and in addition to lipid-lowering effects, suppression of obesity and diabetes has been observed. Therefore, the title of this paper, Inhibition of Hyperlipidemia and Oxidative Stress, needs to be revised.

Response: Response: We thank the reviewer for this insightful comment. We agree with this comment therefore we have followed the suggestion and changed the title to reflect the reviewer’s suggestion. The title is now ‘Doenjang Ameliorates Diet-Induced Hyperlipidemia and Hepatic Oxidative Damage by Improving Lipid Metabolism, Oxidative Stress and Inflammation in ICR Mice’.

  1. Experimental Data: DS altered several indices very strongly. For example, in Figure 4, DS intensely inhibited NF-kB and IL-6 mRNA expression to below normal levels. In Figure 5, it also suppressed ACC and FAS expression, which are involved in lipid synthesis and metabolism, to almost zero. Given these data, such changes observed with high doses of DS should be considered as side effects rather than an effect. Alternatively, it may be related to the lack of essential nutrients mentioned earlier. In Figure 3, DS markedly increased pancreatic Glut4 mRNA expression. The authors need to explain the reason for this. In this study, DS was evaluated in high-fat-fed mice, but what would be expected if DS were given to normal mice? If necessary, the study should be conducted in normal mice.

Response: we appreciate your insightful comment, However, we too were surprised by the significant modulation by doenjang on the aforementioned markers and we ran the qRT-PCR in duplicates and got this result, this may indicate that while the DS mice might totally limit the FAS-ACC pathway of lipogenesis in liver, it might not be the case in other tissues such as the adipocyte tissues. This will be further researched on in future studies. However, in order to confirm this alteration was not due to side effects, we can look to the AST and ALT results of the DS mice which were significantly lower than HD mice indicating lower hepatotoxicity and liver damage. That means intake of 30% Doenjang added to HD did not result in liver damage or stress response. Furthermore, Elevated Glut4 mRNA expression in the pancreas may signify an adaptive response to increased glucose levels or insulin demand as a result of high fat diet feeding. This upregulation could facilitate enhanced glucose uptake into pancreatic β-cells, leading to improved insulin secretion in response to elevated blood glucose levels. Therefore, increased pancreatic Glut4 mRNA expression may indicate improved pancreatic function and glucose homeostasis, potentially serving as a marker for enhanced pancreatic β-cell function and insulin sensitivity. This supposition/hypothesis is consistent with our study findings. We have added this information in the revised discussion. As regards to a lack of normal mice supplemented with doenjang, this could be considered a limitation of this study, we were more focused on comparison between high fat diet mice groups and incurred this oversight. We shall note this point for future studies.

  1. Ingredient composition of DS: As noted above, there is no clear explanation of which parts of the body DS acts on and how it exhibits obesity and lipid-lowering effects, but another problem is that there is no description of what ingredients are in DS. It appears to be a fermented soybean product with salt added as an ingredient, but it needs to be clarified what ingredients are responsible for these effects. What is the substance that is acting? Is it acting in the intestinal tract or is it absorbed and acting? How much salt does DS contain, and there is a concern that salt may have a significant effect when DS is given in large doses?

Response: Thank you for this observation, In the introduction, we gave a brief explanation about the bioactive ingredients in Doenjang such as soy isoflavones and peptides. As for the ingredients in DS we explained the manufacturing process of doenjang in the methods section under subsection 2.1 Animals and diets which indicates Doenjang contains steamed soybeans which are fermented with fermentation bacteria Aspergillus oryzae and Bacillus subtilis for a period of one month. After which, it was mixed with brine (saltwater) and further fermented for two additional months. The Salt content of traditional doenjang averages at 12% (Park S.K., Seo K.I. Quality Assessment of Commercial Doenjang Prepared by Traditional Method. J. Korean Soc. Food Sci. Nutr. 2000;29:211–217) (Mun EG, Park JE, Cha YS. Effects of Doenjang, a Traditional Korean Soybean Paste, with High-Salt Diet on Blood Pressure in Sprague-Dawley Rats. Nutrients. 2019 Nov 12;11(11):2745. doi: 10.3390/nu11112745. PMID: 31726743; PMCID: PMC6893577). Furthermore, there have been studies regarding the effect of this added salt to doenjang activity. In fact, our laboratory on-going and future study have plans to assess the effect of the salt contained in DS by comparing it to a mice group fed similar salt content added to HD diet. We have termed this line of research as a ‘salt paradox’ of Korean fermented foods. We already did a similar study design for gochujang; another Korean fermented food with salt content (Edward OC, Lee EJ, Han A, et al. Gochujang Consumption Prevents Metabolic Syndrome in a High-Fat Diet Induced Obese Mouse Model. J Med Food. 2023 Apr;26(4):244-254.) and plan to do the same for doenjang as well in the future.

[Minor points]

line 79: HFC may be a mistake for HD

line 79-80: What is Sunchang region? Is DS a mistake?

Response: We have corrected that mistake in line 79 and have changed HFC to HD.

Sunchang region is a region in Korea famous for the production of traditional Korean fermented foods such as gochujang, doenjang, etc. our doenjang sample used in the current study was sourced from that region.

DS was the label we chose for the Doenjang Supplemented group; However, we understand how it could be confusing by placing it in that line, therefore, we have removed ‘DS’ from the aforementioned line for better readability.

Additional clarifications

We have revised our manuscript to address the various problems brought up by the esteemed reviewers and hope we have been able to resolve the reviewers’ concerns and that our manuscript is now acceptable to be published in your esteemed journal. In addition to the above comments, all spelling and grammatical errors pointed out by the reviewers have been corrected.

Sincerely,

Edward Olivet Chiamaka

First author

2024-04-11

Round 2

Reviewer 1 Report

Comments and Suggestions for Authors

The data and all comments were edited except Figure 6 (A), which needs to be improved the name of Protein "GPx1....?, and provide a supplement of the original Western blot data (if any).

Author Response

Thank you for your insightful comment. We apologize for our oversight and have rectified the labeling of Figure 6A to precisely correspond with the associated graphs in Figure 6. To enhance clarity and comprehension, GPx1 and GPx2 have been distinctly separated. Additionally, the labeling of the GPx homotetramer in the graph has been rectified.

Reviewer 2 Report

Comments and Suggestions for Authors

You have answered in detail the questions I have posed. I agree with most of the answers, but I wonder if the percentage of DS added to the feed (30%) is viable as an experimental system. As you have answered, there may be some similar examples in the past, but it would be difficult to accurately evaluate the dietary material in such a system.

Comments on the Quality of English Language

The quality of English is generally good.

Author Response

We acknowledge your insightful comment, we know previous investigations have delved into the health-promoting properties of doenjang, with a lower supplementation percentage of 14.4%. Examples include, Nam et al. (2015) which demonstrated the inhibitory effects of doenjang on oxidative stress and inflammation in adipose tissue, and Ko et al. (2019) showed its potential in ameliorating neuroinflammation and neurodegeneration in mice fed a high-fat diet.

In our endeavor to devise a novel research design, we have decided to incorporate a higher dosage of 30% doenjang into the high-fat diet. This decision stems from a synthesis of our prior investigations, alongside insights gleaned from relevant literature, including studies on supplementation with other traditional fermented foods in the Korean context. For example, Kim et al. (2013) explored the impact of Cheonggukjang intake on metabolic parameters in mice fed a high-fat diet with a supplementation percentage of 30%.

While we acknowledge the reviewer's apprehension regarding the potential for excessive dosage, we believe this still provides new important preclinical data that will be helpful for future researchers. In addition, we will address this concern in forthcoming studies by formulating experimental groups with varying supplementation concentrations, spanning from low to high doses.

  1. Nam YR, Won SB, Chung YS, Kwak CS, Kwon YH. Inhibitory effects of Doenjang, Korean traditional fermented soybean paste, on oxidative stress and inflammation in adipose tissue of mice fed a high-fat diet. Nutr Res Pract. 2015 Jun;9(3):235-41. doi: 10.4162/nrp.2015.9.3.235. Epub 2015 May 6. PMID: 26060534; PMCID: PMC4460054.
  2. Ko JW, Chung YS, Kwak CS, Kwon YH. Doenjang, A Korean Traditional Fermented Soybean Paste, Ameliorates Neuroinflammation and Neurodegeneration in Mice Fed a High-Fat Diet. Nutrients. 2019 Jul 24;11(8):1702. doi: 10.3390/nu11081702. PMID: 31344808; PMCID: PMC6723205.
  3. Kim J, Choi JN, Choi JH, Cha YS, Muthaiya MJ, Lee CH. Effect of fermented soybean product (Cheonggukjang) intake on metabolic parameters in mice fed a high-fat diet. Mol Nutr Food Res. 2013 Oct;57(10):1886-91. doi: 10.1002/mnfr.201200700. Epub 2013 Apr 23. PMID: 23609950